# Sustainability, Innovation and Rural Development: The Case of Parmigiano-Reggiano PDO

**Filippo Arfini** [1,*] , **Federico Antonioli** [2] , **Elena Cozzi** [1] , **Michele Donati** [3] ,
**Marianna Guareschi** [1] , **Maria Cecilia Mancini** [1] **and Mario Veneziani** [1]

1   Department of Economics and Management, University of Parma, Parma 43125, Italy
2   Department of Agriculture and Forestry, University of Tuscia, Viterbo 01100, Italy
3   Department of Chemistry, Life Sciences and Environmental Sustainability, University of Parma,
    Parma 43125, Italy
*   Correspondence: filippo.arfini@unipr.it

**Abstract:** Sustainability is becoming a pivotal guide for driving the governance strategies of value chains. Sustainable policy should have as its objective the perpetuation of production models over time to maintain its environmental, economic and social dimensions. Therefore, measuring the sustainability of a production system is fundamental to deepening the understanding of ongoing trends, considering the pressure exerted by agricultural policies, market dynamics and innovations introduced in the production system. The purpose of this paper is to present a holistic framework for assessing the sustainability of food quality schemes (FQS), including the role of both stakeholders within the value chain, and the territorial dimension. This paper discusses the use of dimensional indicators and proposes synthetic indexes to provide an overall picture of the evolution of sustainability of a specific production system. Particularly, the evolution of sustainability in the Parmigiano Reggiano Protected Designation of Origin (PDO) production system is evaluated over the period 2000-2018. It is assumed that its evolution is due to the effect of 20 years of innovations which have impacted on product quality, value chain performance and rural development, modifying the sustainability of the whole production system.

**Keywords:** innovation; local agri-food system; rural development; sustainability

## 1. Introduction

Sustainability is a fairly complex concept put forward by the Food and Agriculture Organization of the United Nations (FAO) which detailed its three environmental, social and economic dimensions [1]. Therefore, sustainable production should take into consideration all the inputs (both natural and social resources) that contribute to the creation and the valorization of food products over time. Special attention should be devoted to the protection of natural and social resources, particularly ensuring their long-term reproducibility and, hence, allowing people of rural areas involved in the production systems to continue their activities in their territories. This also represents the objective of the European Union (EU) rural policies that aim to improve the quality of life for all the rural population engaged in both farming and non-farming activities.

Defining sustainable development becomes a more complex task when innovations are introduced, spurring economic growth, or when the latter faces technical or social problems. In this regard, the FAO, with the purpose of promoting sustainable local agricultural products, has developed the concept of the "virtuous quality circle" [2], suggesting the codification of natural resources and production techniques in order to manage the reproduction of the system over time. The impact of innovations is particularly relevant for geographical indications (GIs), which include the Protected Designation of

Origin (PDO) and the Protected Geographical Indication (PGI) of the EU, since they act both on the value chain and on the territory where the inputs are produced and processed [3,4]. The inputs usually present specific quality features, while the processing phase reflects the culture of the producers on the basis of their capacity to adapt their process to the local environment [5]. The value chain concept represents not only the evolution and the trajectory of a product, but embodies the complex system of relationships among agents from production to consumption. The concept of a value chain combines the technological functions of the supply chain in wider economic and managerial actions. For the agri-food sector in particular, the value chain is regarded as a production management tool useful to create defined product quality levels and to develop marketing strategies aimed at creating value for all the actors of the chain [6].

The structural characteristics and the dynamics of the value chain, however, are not sufficient to assess the impact on the sustainability of the GIs production systems, whose efficiency is the result of the embedding condition between the value chain and the territory that gives the name to the GI of interest. A theoretical framework can be defined to interpret the environmental-social-economic domains where the actors and stakeholders develop their strategies for producing, trading and consuming the GI products. This is instrumental to a better understanding of the determinants of the sustainability of the GIs systems.

The scientific debate around the role of the territory in terms of its contribution in enhancing the level of economic competitiveness often recognises the industrial district (ID) [7] as the most efficient industrial organization model. The ID offers a model of production that can help small and medium sized enterprises (SMEs) to attain the same level of competitiveness as large firms and, thus, contribute to economic growth and social development [8]. The same concept is also useful to observe and evaluate the sustainability of the production system that, in turn, coincides with the territory.

A useful conceptualization of the interaction between the territory and the value chain is the local agri-food system (LAFS) [9,10]. The LAFS concept is similar to the ID, since it is considered as a multi-dimensional concept able to raise the competitiveness of the territory by forging opportunities with a focus on sustainability. Hence, LAFSs and IDs represent models of economic growth, social development and environmental management. Their main characteristics are the linkage with the territory in all its dimensions, also including the role played by all the typologies of territorial agents (i.e., environmental, economic and social), and their institutions managing marketing strategies, local resources and specific environmental characteristics. Three distinctive features identify a LAFS:

i.　The place: Intended in its broadest meaning, also known as "*terroir*", it covers the specific nature of natural resources, the production history and tradition, and the presence of local know-how [11–14];

ii.　The social relationships: These consist of trust, reciprocity and co-operation among actors. They are the glue of local action and an endogenous development mechanism can arise from their interaction the with the place [9];

iii.　The institutions: Private and public agents who promote actions regulated by formal and informal rules [8].

The LAFSs can take different forms depending on the role that the natural environment, agricultural sector and food industries have in the production process and in managing the whole system [3]. The way in which agri-food systems reorganize themselves, meet consumer needs, generate positive (or negative) externalities and trigger spatial dynamics, is a cause rather than an effect of the evolution process. The interaction among LAFS stakeholders is then a central point when defining the evolution of the local system, considering the link between the territory and the food chain. The scheme of possible combinations between food chains and territories leads to different typologies/classes of LAFSs [3]:

i.　The closed system: The local agricultural outputs are processed by local food industries (mainly SMEs), and purchased by local consumers. This typology is characterized by a strong and unique

link between agricultural production and the processing phase, companies and/or the local consumers with a great impact on the product quality, firm structure, market strategies and the relationships with the environment.

ii.   The open system: The agricultural outputs are not processed by local food industries or purchased by local consumers. This typology is characterized by value chains where the upstream and downstream actors may not belong to the territory. This happens whenever the local supply does not satisfy the demand of the input from processors and when local consumption is not able to completely absorb the output inducing the LAFS to look for larger markets. Moreover, in open LAFS models, local companies might benefit from connections with local and non-local research systems, which allows them to innovate and follow new technological paths, raising their competitiveness without losing the link with local traditions.

iii.  The mixed system: This is a coexistence of close and open LAFS. This system is characterized by the coexistence of both closed and open LAFS models. The territory at the same time has specific natural characteristics and develops strategies that are typical of both industrial and rural districts with the co-existence of the industrial and rural productions and the industrial and rural externalities. The outcome of this combination is the reinforcement of all the variables that characterize and influence the development process of local areas, including reputation. Nevertheless, problems can emerge among producers if they have different strategies and views on the use of local natural resources [15].

When the LAFS includes the value chain and presents the feature of the closed production system, the local environment is the most critical aspect since the reproduction of natural resources and reinforcing the image and the reputation of the entire system contributes to producing inputs and the volume of production at the specific quality level. The characteristics of local resources become then relevant since they are not just linked to environmental characteristics (e.g., land and water), but also to those aspects, like biodiversity, animal breeds, and local tradition, with highly specific features associated with the history and the natural environmental conditions of the region. Their specificity, thus, is in contrast with the standardized resources, which are generic and reproducible by definition, and characterizes the quality of the final product and contributes to defining the local food quality [16,17].

The LAFS seems appropriate for the analysis of sustainability as it might include several different dimensions of the production and consumption patterns in relation to the specificity of the production system under investigation. As discussed earlier, the LAFS can be referred to as the only production and/or industrial transformation phase that assumes different forms according to the target market, that is whether it is represented by local consumers as it occurs in short food supply chains (SFSC), or distant consumers [3,4]. The LAFS represents the framework in which both dimensions of the production and consumption (no matter if local) of food meet. In relation to the production or consumption strategy, they generate impacts on the economic, social and environmental sustainability.

Moreover, when agri-food systems generate public goods, all the sustainable dimensions of the LAFS become part of the territorial asset [10,18] since the quality of food is closely linked to the quality of both the environment and social relationships among actors. Hence, the LAFS becomes a suitable dimension for interpreting economic changes and strategies within a rural community of citizens and entrepreneurs involved in a process of cumulative knowledge. Economic actors specialize in the production of certain types of goods (or services), satisfying the needs (or desires) of consumers inside and outside the local area, following the logic of sustainable development. However, unlike local development, rural development includes natural resources as active components of the production systems. The evolution of natural resources should be carefully managed to avoid future drawbacks related to environmental issues, volume of production, quality and sustainability of the whole system.

In conclusion, the enhancement of local products through the activation and capitalization of tangible and intangible assets, which include social capital and natural resources, may allow a fair remuneration and, therefore, the re-production of the LAFS by encouraging the preservation of the

territorial system with regards to the social, economic and environmental dynamics. On the contrary, inadequate remuneration of local resources, especially labor, negatively impacts on the production systems by modifying the technologies, increasing the human pressure, reducing the intrinsic quality of the final products and the reproducibility of the system.

It becomes clear how the sustainability of GIs systems depends on a close relationship between the value chains and territorial institutions. The link between the two institutions, managing the value chain and the territory, respectively, guarantees its sustainability, acting on both the environment and cultural and social dimensions combined together. It is possible to argue that there is a cause-effect relationship between the strategy applied by the LAFS actors and the impact on the economic, environmental and social sustainability variables of their decisions.

The evolution of GIs systems is then related to all the elements that are the result of the governance process at a corporate, collective and policy level, including innovation. This latter, especially in LAFSs, changes the relationships among local inputs and the sustainability of the whole system. Indeed, an ex-ante analysis is essential in order to catch and describe the impacts over all the dimensions of sustainability, both in the value chain and in the area of production.

Regarding all of these aspects, the assessment of the impact of innovation on the sustainability of GIs systems requires: i) A holistic approach for the assessment of sustainability, which includes the definition of the indicators and a methodology for their normalization; ii) the definition of the area of analysis (the LAFS); iii) the definition of innovations that put the sustainability of the LAFS under pressure.

The objectives of this paper are to develop a holistic framework allowing the assessment of the sustainability of food quality schemes (FQS), including the role of both the stakeholders within the value chain, and the territorial dimension. This paper discusses the use of dimensional indicators, and proposes synthetic indexes to provide an overall picture of the evolution of sustainability of a specific production system. In particular, the evolution of sustainability in the Parmigiano Reggiano PDO production system is assessed, considering the years 2000 and 2018. The paper is organized as follows: i) The description of the methodology; ii) the description of the innovations that were introduced over the period 2000–2018; iii) the assessment of the sustainability level as captured by the indicators; and iv) the discussion of the synthetic results.

## 2. Materials and Methods

### 2.1. The Theoretical Framework to Assess the Impact of Innovation on LAFS

The theoretical framework adopted in the present research follows the LAFS approach which enables the analysis of the evolution of the main socio-economic and environmental variables under market and innovation pressures, both at the value chain and territorial level. The analysis consists of three steps: i) The definition of the variables able to catch the evolution of the sustainability of the production system; ii) the analysis of their evolution over time; iii) the analysis of the motivations that explain their evolution.

A list of indicators aiming to describe the impact on sustainability was developed in the EU Horizon 2020 project Strength2food (S2F), (the project "Strenght2Food: Food quality for sustainability and health" is funded by the European Union's Horizon 2020 research and innovation programme under grant agreement No 678024), starting from the approach proposed by FAO in the Sustainability Assessment of Food and Agriculture (SAFA) system of indicators. The SAFA indicators aims to describe the economic, social, governance-related, and environmental features of the agricultural and food systems, with a list of over 100 indicators computed on a self-assessment basis. The SAFA indicators concern 21 themes and 58 sub-themes covering the four above-mentioned dimensions [18]. The S2F project used 23 indicators representing the contribution to sustainable development in the environmental, social and economic dimensions of different food systems [19]. Unlike the SAFA method, which considers the food system as the value chain, the S2F one focuses on the LAFS, which,

in turn, accounts for the contribution of the value chain and the territory of production. For this reason, the S2F method uses indicators which are defined and computed both at the chain level (farm and at processing stages) and at the territorial level. This latter coincides with the GI region. This approach provides an indirect measure of the sustainability of the entire production system, considering territorial variables strictly connected to the respective food productions.

All of the indicators used in the S2F project are the result of a specific elaboration [20,21] based on the use of primary data, specific to the case study and found by field research, and secondary data from available databases.

As in the SAFA philosophy, sustainability was identified and defined according to three classes of externalities (environmental, social and economic), assuming that each innovation generates an impact on one (or more) of these three dimensions. The main assumption is that corporate and governance decisions aimed at improving socio-economic phenomena (as market evolution or the introduction of innovations) generate a cross-cutting-effect on all the three sustainability dimensions. Thus, their impact can be observed by defining a baseline and tracing its evolution over time. This paper focuses only on the province of Parma (which the province in which the greatest volume of Parmigiano Reggiano PDO cheese is produced out of all the provinces included in the production area), analyzing both the effects on the chain and on the rural area, which is differentiated by altitude.

The data needed to describe the state of sustainability of the LAFS has different sources. The data referred to the value chain comes from the activities undertaken in the S2F Project, while those related to the regional analysis come from secondary data collected for administrative and statistical reasons. For the purpose of this analysis only, a set of the indicators computed in the S2F project were used. Moreover, other indicators that aim to describe the structure of the agricultural supply chain (farms and dairies) and rural areas, differentiated by altitude in the years 2000 and 2018, were added. The set of indicators in Table 1 includes: variables which are common for all the agents, while others are differentiated by altitude; the variables which refer to the value chain dimension and others which refer to the territorial dimension.

The aim of these variables is to describe the main economic, social and environmental impacts that are observed at different levels of the LAFS. The economic and social information for the year 2000 and 2018 are drawn from several sources available at the NUTS 4 level of the classification of EU regions such as: i) the Italian FADN (for the economic information); ii) the register of milk producers managed by the Emilia Romagna Region (for structural information); iii) the statistical portal of the Province of Parma (for all the social information) [22]. The environmental information was obtained from the S2F project [23]. The goal of the value chain information is to describe the state of the value chain with respect to sustainability, while the goal of the territorial information is to describe the state of different rural areas (in this case, the Province of Parma). The use of the variables in two moments in time enables the consideration of the evolution of LAFS characteristics, such as the population, markets, agricultural and social policies as well as the introduction of innovations. As detailed in Table 1, all the variables reflect the LAFS features. Some variables are related to the value chain and others to the territory. These latter are also differentiated by altitude (plain, hill, mountain). In total, 20 variable typologies were used.

The economic variables aim to catch the main significant information that describes the economic and structural feature of the LAFS (price, value-added, operating margin, farm and dairy structure, farm production capacity, farm productivity). One interesting economic indicator is the Local Multiplier3 (LM3), which is the only economic indicator at the territorial level which enables the calculation of the local economic impact of the dairies operating in a given local area [19], and more specifically, the amount of Euros remaining in the local area for each Euro of output sold on the market.

The environmental variables are focusing only on two aspects: The use of water and the pressure of the farming system due to the intensity of its breeding system. On the contrary, the social indicators are referred mostly to the territorial level. They catch how the local population evolves with respect to the anthropic pressure, the work opportunities and the capacity to generate/attract youth in their

municipality, and the social role of the LAFS in aggregating the supply of milk by farmers to a single dairy (especially if it is a cooperative dairy).

**Table 1.** Variables for the assessment of the impacts of sustainability.

| Type of Sustainability | Variable Typologies | Value Chain and Territorial Differentiation | Altitude Differentiation |
|---|---|---|---|
| Economic | Price | | None |
| | Gross Value-Added | | |
| | Gross Margin | | |
| | Agricultural Structure | | |
| | Productive Structure | Value chain | |
| | Production Capacity | | Plain/hill/mountain |
| | Milk Productivity | | |
| | Work Productivity | | |
| | Industrial Structure | | |
| | Local Multiplier | Territorial | |
| Environment | Green Water Footprint (Net Consumption of Water) | | None |
| | Grey Water Footprint (Water Pollution) | Value chain | |
| | Blue Water Footprint (Gross Consumption of Water) | | |
| | Production Pressure | | |
| | Anthropic Pressure | | |
| Social | Total Employment | Territorial | Plain/hill/mountain |
| | Industrial Employment | | |
| | Agricultural Employment | | |
| | Senility | | |
| | Social Aggregation | Value chain | |

Source: authors' elaboration.

### 2.2. How to Assess Different Sustainability Dimensions in LAFS

The objective of the use of indicators is to set benchmarks that can be updated in order to provide useful information to agents and stakeholders. It assists them in managing both the value chain and the rural policy, to produce externalities and to reproduce the Parmigiano Reggiano PDO LAFS.

Each indicator reports the different impacts and can be shown individually (as SAFA and S2F suggest) in a multi criteria logic or can be combined into a single composite indicator. Both approaches present advantages and disadvantages in relation to the objectives of the analysis. To fulfil the aim of this research, the approach of the composite indicator is used, which allows the representation of a synthesis of the dimensions of sustainability, facilitating the evaluation by non-expert and policy makers.

The composite indicators are, by definition, multi-dimensional and are intended to describe a complex system of different phenomena captured by single dimensional indexes. The aggregation of different indexes and dimensions undertaken here is similar to the one adopted by the United Nations Development Program (UNDP) in computing the Human Development Index (HDI), which combines the dimension of a long and healthy life with the access to knowledge, and a decent standard of living. In the present study, the challenge was how to treat and calculate a "comprehensive" sustainability index which aggregates single indicators representing the different sustainability dimensions. For the purpose of this research, the definition of composite indicators of sustainability, the following steps are adopted:

i. The definition of dimensional indexes: They are the indicators which report the observed value with respect to the deviation from the values of other homogeneous observations. It is calculated as: Dimensional index = (actual value−minimum value)/ (maximum value −minimum value) (1).

The dimensional indexes are normalized using a quantitative scale from 0 to 10, where 0 represents the lowest level (i.e., the lowest impact on sustainability) and 10 the highest. The normalization aimed to obtain comparable indexes (pure numbers), on the one hand, and to summarize them in aggregate indexes, on the other hand. To pursue the first aim, the indicators were simplified and grouped into environmental, social and economic.

ii.   The definition of an aggregate synthetic index through the calculation of the geometric mean of the dimensional indexes: Synthetic index = (Dimensional index 1 × Dimensional index 2 × Dimensional index 3)$^{1/3}$

The number of dimensional indexes can vary according the objective of the analysis and the dimension of the phenomena that should be analyzed. The synthetic index may have originated from the dimensional indexes for a single dimension of sustainability (i.e., aggregating the indicators for each dimension into the respective dimensional indexes first, and then aggregating the environmental, social and economic dimensional indexes into the synthetic index) or aggregating all the indicators at once into the synthetic index.

The extant literature reports on several methods of weighting and aggregating indicators according to the purposes, the scales, and the perspective adopted [24]. For the purpose of this research, the method adopted was aggregation through a geometric mean. A relevant feature of this aggregation method is that small values are much more influential than in the case of employing an arithmetic mean. Furthermore, the presence of a single null value of a dimensional index is sufficient to yield a zero value of the synthetic index. This last feature gives the dimensional index a strong meaning since if it is null, the system is not sustainable.

This is in line with the purpose of assessing the state of a particular production, as pointed out by Gan et al. [24], although a strong sustainability perspective is also relied upon [25,26]. The choice of relying on a strong perspective, rather than on the weak one, reflects the idea that all dimensions contribute equally to generating sustainability. It further places this study in line with those which take into account other sustainability dimensions besides the purely economic one [26].

Consequently, in this research, after a focus group with representatives of the stakeholders on the observed values (reported in Table 2), the equal weighting of the indexes was employed. This study thus proceeded by computing the dimensional indexes (presented in Figure 1) and the synthetic indexes (presented in Table 3) for each sustainability dimension at the farm and processing level.

**Table 2.** The observed value used to represent sustainable indicators per year and value chain level.

| Indicators | Name | Sign | Unit | Value at Farm Level | | Value at Dairy Level | |
|---|---|---|---|---|---|---|---|
| | | | | Year 2000 | Year 2018 | Year 2000 | Year 2018 |
| **Economic indicators** | | | | | | | |
| | Local multiplier | + | | 2.40 | 2.50 | 2.40 | 2.50 |
| | Price | + | Euros kg-1 | 0.40 | 0.50 | 9.00 | 9.30 |
| | Gross value-added | + | % of turnover | 65.00 | 54.90 | 7.30 | 7.80 |
| | Gross operating margin | + | % of turnover | 63.00 | 52.50 | 2.30 | 2.60 |
| Hill | Agricultural structure | + | Ha/farm | 12.70 | 18.10 | | |
| Mountain | | | | 8.90 | 13.80 | | |
| Plain | | | | 14.50 | 21.60 | | |
| Hill | Productive structure | + | Cows/farm | 38.70 | 84.10 | | |
| Mountain | | | | 21.50 | 51.50 | | |
| Plain | | | | 49.50 | 99.00 | | |
| Hill | Production capacity | + | Milk produced (t)/farm | 228.50 | 498.20 | | |
| Mountain | | | | 124.90 | 304.40 | | |
| Plain | | | | 296.90 | 591.80 | | |
| Hill | Milk productivity | + | Milk produced (t)/cow | 5.80 | 6.00 | | |
| Mountain | | | | 5.70 | 5.80 | | |
| Plain | | | | 6.00 | 6.00 | | |
| Hill | Work productivity | + | Milk produced (t)/Annual Working Unit (AWU) | 1.20 | 2.10 | | |
| Mountain | | | | 3.10 | 6.50 | | |
| Plain | | | | 1.50 | 2.70 | | |
| Hill | Industrial structure | + | Processed milk (t)/dairy | | | 2.10 | 2.51 |
| Mountain | | | | | | 2.32 | 2.50 |
| Plain | | | | | | 1.78 | 2.56 |
| **Environmental indicators** | | | | | | | |
| | Green water footprint (net consumption of water) | − | m3 kg-1 | 4.80 | 4.30 | | |
| | Grey water footprint (water pollution) | − | m3 kg-1 | 0.60 | 0.50 | | |
| | Blue water footprint (gross consumption of water) | − | m3 kg-1 | 8.80 | 7.33 | 58.00 | 51.50 |
| Hill | Production pressure | − | N. cows/ha (Utilized Agricultural Area) | 0.60 | 0.70 | | |
| Mountain | | | | 0.80 | 0.40 | | |
| Plain | | | | 0.80 | 0.80 | | |
| **Social indicators** | | | | | | | |
| Hill | Anthropic pressure | + | Inhabitants/Km2 | 139.10 | 165.20 | 139.10 | 165.20 |
| Mountain | | | | 22.40 | 20.70 | 22.40 | 20.70 |
| Plain | | | | 167.40 | 202.00 | 167.40 | 202.00 |
| Hill | Total employment | + | % of inhabitants | 45.60 | 45.40 | 45.60 | 45.40 |
| Mountain | | | | 37.10 | 41.70 | 37.10 | 41.70 |
| Plain | | | | 45.70 | 43.40 | 45.70 | 43.40 |
| Hill | Industrial employment | + | % of inhabitants | 20.60 | 17.50 | 20.60 | 17.50 |
| Mountain | | | | 15.80 | 15.50 | 15.80 | 15.50 |
| Plain | | | | 20.40 | 16.90 | 20.40 | 16.90 |
| Hill | Agricultural employment | + | % of inhabitants | 2.80 | 3.00 | 2.80 | 3.00 |
| Mountain | | | | 3.50 | 3.90 | 3.50 | 3.90 |
| Plain | | | | 3.60 | 3.30 | 3.60 | 3.30 |
| Hill | Senility | − | % of inhabitants | 181.80 | 159.30 | 181.80 | 159.30 |
| Mountain | | | | 416.20 | 408.20 | 416.20 | 408.20 |
| Plain | | | | 178.00 | 155.50 | 178.00 | 155.50 |
| Hill | Social aggregation | + | Farms/dairy | | | 9.40 | 6.00 |
| Mountain | | | | | | 18.60 | 9.40 |
| Plain | | | | | | 6.00 | 4.30 |

Source: S2F and authors' elaboration.

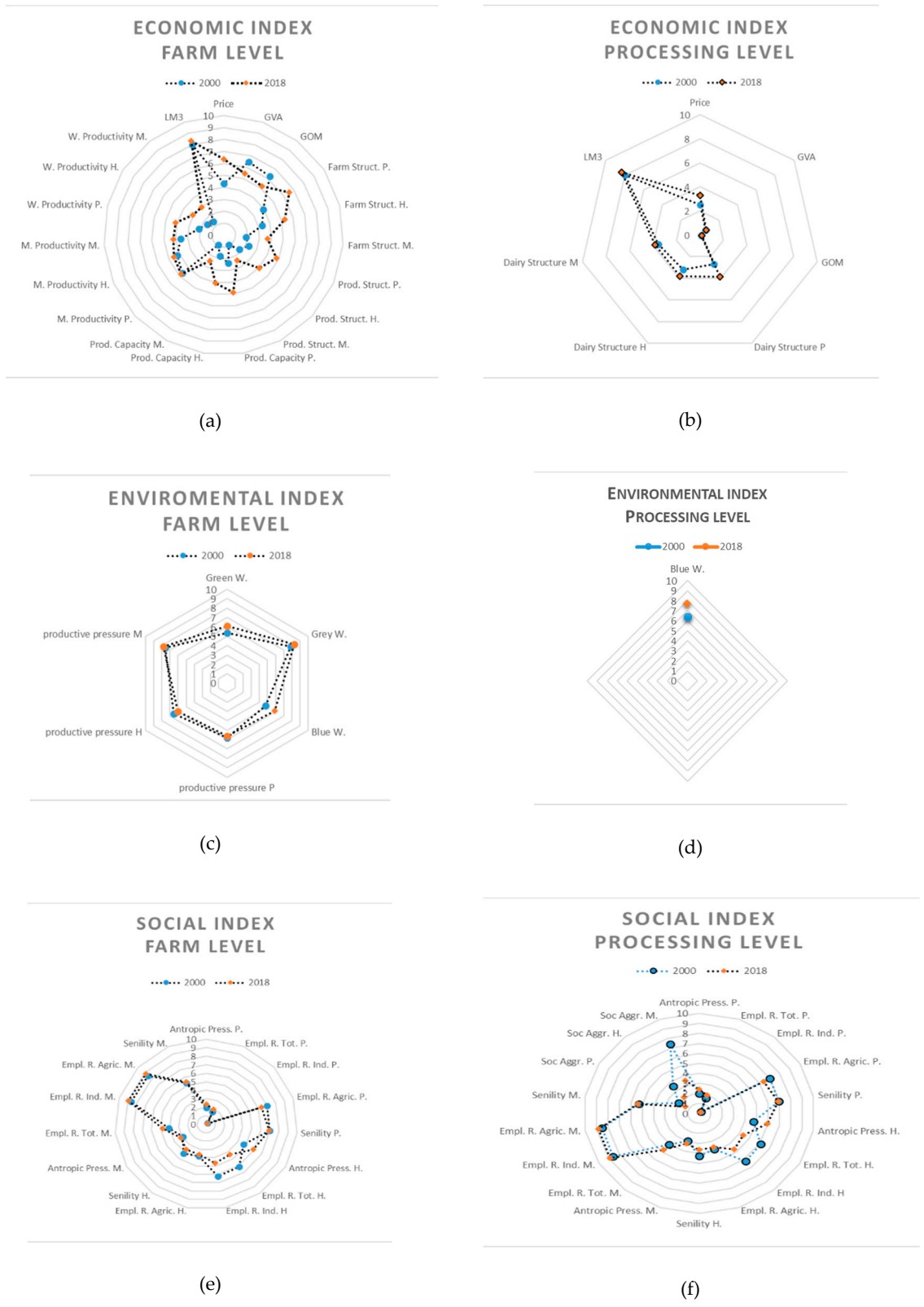

**Figure 1.** Dimensional indexes per year and value chain level. (**a**), economic index farm level; (**b**), economic index processing level; (**c**), environmental index farm level; (**d**), environmental index processing level; (**e**) social index farm level; (**f**), social index processing level. Source: authors' elaboration.

**Table 3.** Synthetic sustainable indexes per year.

| Synthetic Indexes | Farm Level | | Processing Level | |
|---|---|---|---|---|
| | 2000 | 2018 | 2000 | 2018 |
| Global Synthetic index | 3.2 | 4.0 | 2.6 | 2.5 |
| Economic Synthetic index | 2.7 | 4.4 | 1.7 | 2.0 |
| Environmental Synthetic index | 6.2 | 6.6 | 6.4 | 7.7 |
| Social Synthetic index | 3.7 | 3.7 | 3.7 | 3.4 |

Source: authors' elaboration.

## 3. Results

The calculation of the sustainability indicators of the Parmigiano Reggiano PDO LAFS of the Parma province was conducted employing data for the year 2000 and 2018. Parmigiano Reggiano PDO is one of the best-known Italian PDO cheeses in the world. Its quality depends on a strict code of practice, which regulates milk production and its processing into cheese in a defined production area (five provinces across the Emilia-Romagna and Lombardy regions) as well as the ripening system and the use of logos in commercial activities. In the Parmigiano Reggiano PDO LAFS, natural factors play a central role in typifying the final product. The protection and careful management of the natural resources are important in enabling the survival of the uniqueness of the product. For this reason, alfalfa still today is a substantial proportion of the diet of dairy cows.

The cultivation and the use of this forage guarantees a good level of animal welfare and impacts positively on natural resources as well as landscape maintenance. The Parmigiano Reggiano PDO cheese is still considered an artisanal product since the cheesemaker still plays a fundamental role in defining the quality through his knowledge and ability to manage milk produced (potentially) in different natural and managerial contexts. Moreover, the social role of Parmigiano Reggiano PDO is fundamental since it requires aggregating the supply of milk by several farmers to a single dairy. They are not only suppliers, but through the Parmigiano Reggiano PDO governance system, they become also managers and create strong social links among producers that often are spatially isolated in rural areas. This analysis concerns only the Province of Parma that, nevertheless, in the 2018 production year represents the area with the highest number of dairies (150 out of 330 dairies) and the highest number of cheese-wheels produced (1,286,000 out of 3,699,000 cheese-wheels) [27].

For the aim of this research, 20 variables were exploited, which considering the differentiation for altimetric regions and the availability of some of them for both years led to the possibility of employing 45 datapoints. Most of them, in the domain of environmental, social and economic sustainability, were assessed within the S2F Project. Others were specifically elaborated for the purpose of this research. Reflecting the approach described above, the assessment by synthetic indexes is presented in Table 2.

The description of the phenomena detected by the observed economic, environmental and social indicators gives a rather complex image of the evolution of sustainability due to the increase or decline in the values between the year 2000 and 2018. Considering the economic indicators, for example, it is observed that the gross value-added and the gross operating margin decrease for the agricultural companies (even if in modest measure) while they increase for the dairies. This phenomenon is linked to the reduction of the commercial weight of cooperative dairies which by their nature must transfer the profit to the farms, and to the increase in non-cooperative dairies which, on the other hand, produce a profit for the company. Farms have reacted with a classic economy of scale strategy increasing both the structure (ha/farm) and the production capacity (cows/farm). Similarly, dairies have also adopted a scale strategy, significantly increasing their industrial structure (processed milk/dairy). These phenomena reflect on the environmental dimension not so much in the use of water (which is reduced), but in the productive pressure (cows/ha) which increases in the plains and, above all, in the hills. The latter display a greater dynamism than the plain and mountain areas. On the other hand, the mountain area is the area with the most critical evolution of social indicators and manages to guarantee

employment only due to the use of services. However, the most alarming aspect is the sharp reduction in the indicators relating to the social aggregation of the value chain of Parmigiano Reggiano PDO (farms/dairy) which in the mountains are reduced by 50% (from 16.6 to 9.4).

The use of dimensional indicators facilitates reading and interpreting the phenomena that affect the overall sustainability of the system. Figure 1, based on dimensional indices, may assist in appreciating the outcomes of the research effort.

As shown in Figure 1, the representation of the normalized values of the indicators provides an immediate interpretation of the evolution of the production system from a LAFS perspective. This highlights the evolution of the variables that directly refer to the value chain with respect to those that describe the characteristics of the rural area. In the case of the Parmigiano Reggiano PDO LAFS, it clearly shows how sustainability is due to the incidence of economic and environmental variables, to the detriment of social variables. The aggregation through the logarithmic scale, as suggested by Gan et al. [24] and Stiglitz et al. [26], provides an even more synthetic perception of the evolution of the system, indicating how the overall economic and environmental sustainability have improved, but not social sustainability (Table 3). The synthetic indexes score varies from 1 to 10, where 0 is the lowest and 10 the highest level of sustainability.

## 4. Discussion

*Sustainabilty and Innovation Process in the Parmigiano Reggiano PDO LAFS*

The evolution of sustainability in the Parmigiano Reggiano PDO LAFS is quite evident, but the reasons are several and complex. The LAFS is not a static system, but it is subject to internal and external pressures. Usually, the evolution of a system is considered in light of the changes in market conditions driven by the evolution of input and output prices. However, although the Parmigiano Reggiano PDO LAFS is highly regulated through the code of practice, technical progress also contributes to modify and influence the evolution of the system by acting directly and indirectly on the sustainability of the value chain and of the production territory. Somehow, technical progress acts in parallel with the evolution of the market and rural policies, becoming together with the governance of the value chain, one of the tools to improve business resilience. Selected innovations enter in the Parmigiano Reggiano PDO LAFS directly from the market without any filter, while for others, the Parmigiano Reggiano PDO Consortium carries out its regulatory action by introducing rules that regulate their use.

More in general, the issue of innovation, considered both in a broad sense and addressed by the system of GIs, is particularly relevant [14,28,29]. It implies changing the rules between producers, potentially favoring someone and disadvantaging others, and some institutional steps at national and EU level related to the change in production regulations. It follows that in the GI system, the introduction of innovations (generally of an exogenous nature to GIs) is potentially the cause of conflicts between the agents of the value chain. The GIs Consortia intervene by regulating their use and acting directly or indirectly on the potential impact on the production system [29] of the LAFS. As a consequence, the relationship between innovation and the LAFS can be organized in relation to several levels of analysis and potential impacts to achieve a better understanding: i) The innovation and the GI value chain in which the value chain is made up of operators who manufacture the GI; ii) the innovation and consumer perception; iii) the innovation and territory; and iv) the innovation in governance action in GI-LAFS [29].

Assessing the impacts of each individual innovation becomes extremely difficult as innovations often act simultaneously on multiple dimensions of sustainability and on multiple levels of the value chain. For this reason, in this research, the aggregate effect of the introduction of the innovations and their appropriateness with respect to the proposed objectives, measured by the evolution of the sustainability of the Parmigiano Reggiano PDO LAFS, are considered.

In this respect, the Parmigiano Reggiano PDO LAFS offers an interesting case study as: i) The Parmigiano Reggiano PDO cheese (and its territory) is a product with a long history but

with a slow and progressive evolution; ii) as GI the Parmigiano Reggiano PDO is tightly regulated in the technical and managerial aspects through three different internal regulations for milk production, cheese production and the use of marks; and iii) the GI Consortium carries out a strong governance action on the entire production system including the adoption of innovations.

Innovations in the Parmigiano Reggiano PDO LAFS have occurred since the Middle Ages but were largely informal until the 19th century. Only from 1861, following the unification of Italy in a single state and the increase in cheese trade, the Italian government adopted formal rules aimed to protect intellectual property rights, traders and producers, avoiding frauds and bad practices. When internal customs were removed, Parmigiano Reggiano PDO producers faced problems in adhering to the high product quality standards. At that time, eleven small dairies from Bibbiano (Reggio Emilia) producing high-quality cheeses, resulting from cows eating forage from good pastures and from the skill of local cheesemakers, started to show their wares at exhibitions and trade fairs in Italy and abroad [30]. At the end of the 19th century, the Animal and Cheese-making School of Reggio Emilia innovated the production process using the whey to cope with cheese quality problems [29]. This innovation is still employed and allows producing the cheese without relying on preservatives and needing to pasteurize the milk.

Considering the *sui generis* era (just before the introduction of the Reg. 2081/1992 until today), at least 34 different innovations were introduced (Table 4) and are impacting the LAFS (some innovation impact simultaneously on different levels). This shows, in principle, that the EU Regulations on GIs do not stop technical progress. Producers decide which innovation should be adopted in light of the capacity to respect some rules that are considered fundamental. In the case of Parmigiano Reggiano PDO, the golden rule is to produce cheese without any type of preservative other than salt. All the innovations must respect this fundamental feature and, vice versa, innovations that require the use of preservatives in the production of cheese are not allowed.

For the purpose of this research, to tentatively organize the innovations implemented in the Parmigiano Reggiano PDO LAFS, the focus is only on two dimensions: i) the innovation typology—organizational, process technological innovation, product technological innovation; and ii) the innovation impacts—product quality, rural development, value chain competitiveness. This segmentation provides a clear picture of the possible impact that each innovation can generate (Table 4).

Over the period 2000–2018, 21 innovations were introduced in the LAFS (almost 50% of the innovations). This was aimed to increase the quality of the product. They consist, on the one hand, of technological process or product innovations such as hygienic norms or processing rules to assure that the product is safe for human consumption: hygienic regulations to be applied to the processing and transportation equipment; regulations on bacteria content in milk and microbiological and chemical analyses. These innovations limited the incidence of cheese wheels of an unsuitable quality by reducing the bacterial levels in milk. The economic impact is estimated in approximately 300,000 Euros per year [29]. On the other hand, they concern the technological process and the organization of marketing strategies as new packing and consumption models. Vacuum packing of pieces of cheese or packaging sizes for ready-to-eat cheese pieces (e.g., snack portions, cubes, shavings) increased significantly in the period 2013–2015; the number of agreements between the Parmigiano Reggiano PDO Consortium and food companies for co-branding, to meet new types of demand, grew from 150 to 200 between 2015 and 2016 [30–34]. The co-branding of innovative food products for which Parmigiano Reggiano PDO is employed as an ingredient generated additional value added deriving from the synergy between the reputation of the two brands and the taste preferences of two types of consumer [32].

Only four innovations (almost 12%) impact indirectly on rural development. These all are organizational innovations such as the Parmigiano Reggiano PDO Consortium, the PDO designation and regulations that protected local production and processing (e.g., heifers being born within the production area, packaging to occur in the Parmigiano Reggiano PDO production area). The Parmigiano Reggiano PDO Consortium represents a major institutional innovation, with the aim of guaranteeing

the quality of the product, to protect the reputation of Parmigiano Reggiano PDO against fraud and usurpation, and to provide the consumer with confidence in the credence attribute of origin [35]. It introduced, decades before the EU legislation, the objectives and the tools of Regulation 1151/2012 (Quality Package) that gives the product the *ex-officio* protection in the EU and allowing the Consortium to take actions to safeguard producers' incomes [29].

**Table 4.** The number and typology of innovation introduced in the Parmigiano-Reggiano LAFS per potential impact (From 1990 to 2018).

| Type of Innovation | Impacts | | | |
| --- | --- | --- | --- | --- |
| | Product Quality | Rural Development | Value Chain Competitiveness | Total |
| **Organizational Innovation** | **11** | **4** | **7** | **22** |
| Casein plate (traceability) | 1 | | 1 | 2 |
| Delimitation of packaging area | 1 | | | 1 |
| Heifers from production area | | 1 | 1 | 2 |
| ISO 9001 introduced in some cheese dairies | | | 1 | 1 |
| Labelling rules (from 10 to 12 months maturation time) | 1 | | | 1 |
| Maturation temperature rules | 1 | | | 1 |
| Milk payment on quality basis | | | 1 | 1 |
| Milk quota to "milk for PR " quota | 1 | | | 1 |
| Packaging in the PR production area | | 1 | 1 | 2 |
| Product Definition of production area specifications for farms | 1 | | | 1 |
| Product promotion and communication | 1 | 1 | | 2 |
| Product specifications for dairies | 1 | | | 1 |
| Product specifications for maturing and quality check | 1 | | | 1 |
| Protected Designation of Origin (PDO) (1992) | | 1 | 1 | 2 |
| Protection of PDO logo and Consortium brand | | | 1 | 1 |
| Quality segmentation rules by labelling | 1 | | | 1 |
| Third party certification body | 1 | | | 1 |
| **Process Innovation** | **5** | | **7** | **12** |
| Cooling of the storage rooms | 1 | | | 1 |
| Feeding with hay | | | 1 | 1 |
| Mechanical harvesting | | | 1 | 1 |
| Mechanical milking | | | 1 | 1 |
| Milk cooling | 1 | | | 1 |
| Packaging technology spread | | | 1 | 1 |
| Pre-packed grated cheese, portions for snacking, PR as ingredient | 1 | | | 1 |
| Product segmentation | 1 | | 1 | 2 |
| Robot for cheeses cleaning | | | 1 | 1 |
| Spread of lorries with refrigerated milk tanks | 1 | | | 1 |
| Unifeed feeding system | | | 1 | 1 |
| **Product Innovation** | **5** | | **1** | **6** |
| Analysis of preservatives and fat content by certification body | 1 | | | 1 |
| Feed composition | 1 | | 1 | 2 |
| Hygienic norms as to bacteria content in milk | 1 | | | 1 |
| Hygienic norms of equipment for transportation and processing | 1 | | | 1 |
| Microbiological and chemical analysis | 1 | | | 1 |
| Total | 21 | 4 | 15 | 40 |

Source: authors' elaboration from [27].

Finally, 15 innovations (almost 30%) impact directly on the competitiveness of the value chain. They are mainly organizational innovations and technological process innovations. Organizational

innovations are related to: The definition of production areas on farms, the requirement that packaging is carried out within the area of production, milk payments based on quality and traceability, to protect consumers as well as farmers from unfair commercial practices and competition by other farmers; international ISO norms and other retailer quality certifications were adopted to achieve, maintain and communicate a given quality of the product batches, thus facilitating the sale of the cheese; Parmigiano-Reggiano PDO cheese quotas, that have been introduced by the Parmigiano Reggiano PDO Consortium in 2015 at the time of the scrapping of the EU milk quotas, to manage production schedules and, in turn, protecting the value of farm assets via a remunerative price [29]. The technological process innovations, such as mechanical harvesting, mechanical milking, and a robot for cleaning cheese have cut working times and labor force requirements. Others, such as new packing techniques and product formats, allow for a longer shelf life without compromising the core characteristics of the traditional Parmigiano Reggiano PDO cheese [29].

All the innovations interact on different levels of sustainability at the same time, generating positive and negative impacts and acting directly or indirectly on the sustainability of the LAFS. As shown in the previous section of the paper, the evolution of the sustainability of the Parmigiano Reggiano PDO LAFS is quite evident and can be due to the evolution of the innovation path since its main aims was to increase the competitiveness and resilience of the system as well as to optimize the management of the production process along the supply chain in a labor saving logic.

## 5. Conclusions

The analysis of the evolution of the sustainability of the Parmigiano Reggiano PDO LAFS shows different aspects: i) sustainability is a complex concept that requires the synergic use of a set of indicators finalized to catch the direction and the size of the evolution of the system; ii) considering the GIs system, the analysis of the sole value chain does not contemplate the role played by the territories in a sustainable logic. The value chain and territorial dimension should be considered embedded as the sustainability of the value chain affects the sustainability of the territory and vice versa; iii) innovation is a powerful tool to improve competitiveness and resilience, but might have a potential effect on the entire LAFS sustainability and on the process of rural development, that in turn, links production and territory in a development policy.

The methodology employed here to catch and describe the evolution of each sustainable dimension, the impacts and the reasons of such evolution can be of great usefulness to policy makers and entrepreneurs that need to define appropriate strategies and policies. The methodology of multiple normalized indicators provides a picture of the level of sustainability of the Parmigiano Reggiano PDO LAFS which highlights how the effect of innovations, together with the evolution of the markets, and the structure of the production system and the territory, is not neutral.

The main drawback of the proposed methodology is the difficulty in defining the dimensional indicators and finding information at the level of detail suitable to describe the observed phenomena at its best. The collection of primary data can be very costly and difficult. Furthermore, for reasons of privacy, at the same time, also secondary data might present problems especially when the analysis is organized at the NUTS 4 level of the classification of regions. Lastly, this analysis requires a consensus from stakeholders on the definition of the weights for the calculation of the synthetic indicators. The latter represents the weight that stakeholders recognized for the influence of the specific variables on sustainability.

Nevertheless, the proposed methodology, despite some limitations, provides an overall picture of the sustainability of the LAFS. In the case of the Parmigiano Reggiano PDO LAFS, the increased levels in the technological and organizational pressure on the Parmigiano Reggiano PDO system strongly influenced its evolution by changing its characteristics. On a scale of 0 to 10, the synthetic sustainable index in the year 2018 was only 4.0 for farms and 2.5 for dairies. This value is justified by the level of environmental sustainability (above 6 and improving) which is due, above all, to the low production pressure that characterizes the Parmigiano Reggiano PDO LAFS. The index value highlights how the

technological innovations introduced allowed a more sustainable management of natural resources, reducing the negative environmental externalities.

The economic index increases at both the upstream and downstream level. This indicates that technological and process innovations have a positive impact on the value chain, especially at the farm level, whose index increased from 2.7 to 4.4 between 2000 and 2018. Indeed, the Parmigiano Reggiano PDO Consortium policy aimed at supporting farmers' income, protecting their activity and adding value to the raw materials. On the contrary, at the dairy level, a lower economic sustainability emerges since dairies, that are mostly coops, are instrumental in the valorization of milk. Nevertheless, the economic index increased from 1.7 to 2.0 at the dairy level in the same period.

Social sustainability is rather weak in the Parmigiano Reggiano PDO LAFS. This index includes indicators which catch the social evolution of the rural areas. The decreasing value of social sustainability is linked to the social evolution of the rural area and the fact that, on the one hand, dairy farmers have benefited from massive technological labor-saving improvements and, on the other hand, the agricultural sector as a whole is not the main socio-economic activity of the region anymore. The social index remains unchanged at 3.7 in both years at the farm level and declines from 3.7 to 3.4 at the dairy level between 2000 and 2018. Nevertheless, the number of milk farmers per dairy strongly decreased in that period, impacting the aggregated social index in a negative way.

The research on the Parmigiano Reggiano PDO LAFS shows acceptable sustainability conditions between 2000 and 2018 and catch the capacity of the system to react to the increasing technological pressure and market competition. The loss of workers at the farm and dairy levels is the most significant phenomenon undermining the sustainability of the system that, in the long-term perspective, can impact on the image of the product and thus, on the economic sustainability. In a logic of rural development, other job opportunities linked to the Parmigiano Reggiano PDO should be created by the diversification process or specific policies. It is difficult to imagine a return to labor intensive processing techniques along the value chain. However, the LAFS is considered as the synergic activity of dairies and territories that might evolve by offering commercial services (such as direct sales at the dairy level), or recreational touristic services to their customers (consumers and tourists), recovering the loss of jobs from the Parmigiano Reggiano PDO value chain. This last scenario, although desirable, is not easy to achieve and requires the strategic sharing and collaboration of all the stakeholders that manage the LAFS.

**Author Contributions:** Conceptualization, F.A. (Filippo Arfini), E.C., M.G.; Data curation, E.C., M.D., M.G., M.V.; Methodology, F.A. (Filippo Arfini), E.C., M.G.; Software, F.A. (Filippo Arfini), E.C., M.G.; Writing–original draft, F.A. (Filippo Arfini), M.G.; Writing–review & editing, F.A. (Filippo Arfini), F.A. (Federico Antonioli), M.G., M.C.M., M.V.

**Funding:** This paper is one of the outputs of the Srenght2food Project which has received funding from the European Union's Horizon 2020 research and innovation program under grant agreement No 678024.

**Acknowledgments:** The authors wish to thank the Parmigiano Reggiano PDO Consortium for the help received in accessing the relevant data.

**Conflicts of Interest:** The authors declare no conflict of interest.

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
