# Peer review of "Sustainability, Innovation and Rural Development: The Case of Parmigiano-Reggiano PDO"

_sustainability, doi:10.3390/su11184978_

Round 1

Reviewer 1 Report

It is an interesting paper . However, this paper has some problems: it has to clarify the methods, the tables and complete the results with the figures of the table. The tables have to be clarified and arranged  as well described the methodology with more detail. I can see that this paper was made for the research involved in project and they know all variables and indicators but for the ordinnary reader they do not need this detailed information and they have to understand. You have to replace it in order to have a better understanding of the paper. FQS, you should to say what it is, you know but the readers could not.

Please, see my comments on the attached paper

Author Response

Dear Reviewer,

Thanks a lot for your very fruitful comments that I have all considered in the new version of the paper.

After your comments I have modified the structure of the paper. Now the paper focus on the methodology to assess the sustainability of a GI system. The technical progress introduced in the time been 2000-2018 represent one reason of the evolution of the sustainability indicators. So, correctly, the methodology is now focusing on how assess the sustainability, the result section is on the score of the indexes and the innovation is discussed in the “Discussion section”.

Of course I have considered also yours suggestions in the following matter:

I have explained the difference between Industrial and rural district I have clarified the number and the use of indicators I have change the structure of the table and now (I hope) their meaning is more clear I have explained the reason of the geometric mean Now is more clear the link between the methodology and the result I have change title of table 2 (now table 4) and now it is more clear I have clarified the source of the data I have change the name of the variables I have change the label of the Graph. 1

Reviewer 2 Report

This paper is really very difficult to follow which is a shame because it could be an interesting paper. It needs to have a clearer focus, the aim and purpose are not clearly articulated and this might be one of the problems. 

The English is really very poor, there are mistakes in spelling, grammar and punctuation, including incorrect tenses, as well as choice of word e.g. base rather than basis, and capital letters added where not needed.

Mistakes in spelling e.g. Strength2food is spelt at least three different ways, LASF also appears instead of LAFS. Abbreviations need to be in capitals e.g. ‘PDO’ in title not ‘pdo’

You need to explain abbreviations at first use.

The decimal separator in English is a full stop/period and not a comma, thousand separator in English is a comma and not a full stop.

The abstract and title could be more focused to describe more clearly what the paper is about.

Lines 33-34 whose policies, EU policy or Italian policy?

Line 40 – should this be PGI, as in the EU designation?

Lines 63-71 no references for LAFS here, need to justify more clearly why this particular terminology is appropriate and not other terms, e.g. SFSC, AFN, local food etc.

The inclusion of LAFS in this paper is not clearly justified or explained, is the paper looking at only LAFS only closed, or mixed systems Im not sure why is the type of LAFS important to this paper,

Table 1:

Why are the indicators so unbalanced, there are 10 economic indicators, 6 social and only 4 environmental, these seem to miss several key impacts, they don’t take into account biodiversity or GHG emissions for example for the environmental indicators.

Some of the indicators are not obvious from their title, e.g. what does ‘senility’ cover and the difference between ‘blue water footprint’ and ‘green water footprint’. Why is it not ‘population pressure’ rather than ‘anthropic pressure’.

In the Table 1 it should be plain not plane. In this table you divide the last column into value chain or regional, but in the text you talk about territory, need to be consistent.

Lines 270-273 - I guess these relate to Table 5, but the terms used are totally different e.g. text in lines 271-273 describes a ‘general sustainable aggregated index’ in the table this is ‘global synthetic index’.

Table 2:

What are the numbers for that appear to the left of each value?

Table 3:

What do you mean here by ‘processing level’ and ‘farm level’?

In the discussion, you state that sustainability has improved since 2018 however these two snapshots in time do not really allow you to say this, although the 2018 value is greater you are only comparing time points using your data and not a trend.

Table 4 is a figure not a table. What do the numbers mean at the bottom left of each graph?

The figures seem to be all very similar for the two years.

Line 430 - what is SYAL?

Lines 437-439 - how is this more concise?

Lines 449- are these values for the global synthetic index it doesn’t say. Why round them to 1 decimal place (1dp) here when they are to 2dp in the table?

Line 450 - how do you know this is a ‘good’ level of environmental sustainability there is no measure/scale used, no baseline comparison.

Lacks acknowledgement/discussion of the limitations of this approach. I.e. not weighting the indicators, impact of the selection of indicators

Need to base the conclusions on your results, many of these are not based on your data but are rather fanciful.

Author Response

Dear Reviewer,

Thanks a lot for your very fruitful comments that I have all considered in the new version of the paper.

After your comments I have modified the structure of the paper. Now the paper focus on the methodology to assess the sustainability of a GI system. The technical progress introduced in the time been 2000-2018 represent one reason of the evolution of the sustainability indicators. So, correctly, the methodology is now focusing on how assess the sustainability, the result section is on the score of the indexes and the innovation is discussed in the “Discussion section”.

Of course I have considered also yours suggestions in the following matter:

I have completely re-organise the focus and the organisation of the paper. Now is more clear. I will ask for an English review from a native English but I have correct my mistaken. I have correct the decimal separator (sorry) I have rewritten the abstract. I think the title reflect the content of the paper I have explained why LAFS is appropriate framework for this analysis Indicator are unbalanced for the difficulty to provide others data for the variables. Nevertheless the geometric mean used do not require it since do not provide error in the calculation, especially if the assessment is made by each sustainability dimension. CGH was calculated in the framework of the S2F project but the Index coordinator do not allow me, at this stage the publication of the data; meanwhile for the biodiversity…you are totally right and I regret to says that was not calculated for several reason. In principle can be added as indicator (and I wish to do so in the next future). the senility index is a dynamic statistical indicator used in demographic statistics to describe the weight of the elderly population in a given population. It essentially estimates the degree of ageing of a population. It is defined as the coexistence relationship between the elderly population (65 years and over) and the younger population (0-14 years); values above 100 indicate a greater presence of older people than the very young. It is a fairly crude but effective indicator, since in the ageing of a population there is generally an increase in the number of elderly people and at the same time a decrease in the number of younger subjects, and in this way numerator and denominator different in the opposite direction, exalting the effect of an ageing population. The distinction of green, grey and blue water are indicate in the table I would prefer to stay with anthropic, since is a wider concept, but if you prefer I will be happy to change I have change the title of the column and correct mistaken I have completely change the structure of the paper. I would like to demonstrate as using a benchmark is possible to follow the evolution of sustainability and know the reason. Was not my intention to describe a trend Line 450: you are right, but for my perception the index as such, referred to the dimensional index used in the analysis. Conclusion now include the discussion on the limitation and based on the result of our analysis. I have explained abbreviations at first use. I have specified EU policy at lines 33-34

Round 2

Reviewer 1 Report

Congratulations, you are a good work.

Just to be perfect: 

236 - format the table in order to avoid some additional columns, try to redesign or delete the additional columns. 

329 - it is missing a full stop in the final of paragraph.

351 & 354 - do not need a point after 1.

430 - table 4. You must use capital letters : Type of Innovation

Value Chain

Process Innovation 

475 - cut the space before quite

498 - Lastly,

Author Response

Dear Review, 

thank a lo to for your very careful review. 

I have accepted all your suggestion. Best Regards

Filippo Arfini  

Reviewer 2 Report

The amendments undertaken by the authors have improved this article immensely and it is much easier to follow and understand now.

There are still a few minor issues, mostly with the English that need correcting.

Line 27-28- did the FAO first propose this concept, it is usually attributed to another author

Line 34- EU

Line 48 – the value chain concept

Line 167- ‘are’ rather than ‘is’ …. ‘allowing the’ not ‘allowing to’

Line 179-180- …….which enables the analysis of….

Line 211- ‘has’ not ‘have’

Line 212 – ‘comes’ not ‘come’

Line 225 – Environmental information was obtained from the S2F project.

Table 1 needs tidying up, it is rather messy

Lines 251-253 needs rewriting to clarify what the point is

Line 306- strict not severe

Line 351-354 it is normally ‘Figure’ not ‘Graph’

Lines 424-425 need rewriting to make the meaning clear

Line 480 – requires

Line 495 – ……..the difficulties in defining…..

Line 496 – …..and agreement about the level……

Line 500 - …. collectively…..

Line 520- limitations, provides

Line 511 – increases, levels

Line 512 - production

Line 514 - ………supporting farmer’s income, protecting their activity and adding value added to the raw material….

Line 518 – the social dimension of sustainability is the weakest….

Lines 527-8 -this sentence doesn’t read well, it is not clear what this sentence means

Author Response

Dear Reviewer,

I thank you very much for your comments and suggestions.

I have correct the English terms and I have follows all your suggestion: I have modified the table 1 and I have rewritten the sentence that you have indicated. In particular, the conclusion has been rewritten and clarified in several aspects.

Now the paper looks like very different from the previous one. Thank you again. My best regards

Filippo Arfini

Below the new sentence:

Table 1 needs tidying up, it is rather messy: new version:

Type of sustainability

Variable typologies

Value chain and territorial differentiation

Altitude differentiation

Economic

Price

Value chain

None

Gross Value-Added

Gross Margin

Agricultural Structure

Plain/hill/mountain

Productive Structure

Production Capacity

Milk Productivity

Work Productivity

Industrial Structure

Local Multiplier

Territorial

None

Environment

Green Water Footprint (Net Consumption Of Water)

Value chain

Grey Water Footprint (Water Pollution)

Blue Water Footprint (Gross Consumption Of Water)

Production Pressure

Territorial

Plain/hill/mountain

Social

Anthropic Pressure

Total Employment

Industrial Employment

Agricultural Employment

Senility

Social Aggregation

Value chain

Lines 241- and follow needs rewriting to clarify what the point is. The new version is:

The goals of the economic variables aim to catch the main significant information that describes the economic and structural feature of the LAFS (price, value-added, operational margin, farm and dairy structure, farm production capacity, farm productivity) when it is analysed.

Lines 454 and follow…need rewriting to make the meaning clear. The new version is:

Is probably unknown that innovations in the Parmigiano Reggiano PDO LAFS have occurred since the Middle Ages but were largely “informal” until the 19th century. Only from 1861, following the birth of the Italian state and the increase of the cheese trade, the Italian government to adopt “formal” rules aimed to protect intellectual property right, traders and producers, avoiding frauds and bad practices. When internal customs barriers in Italy were removed, Parmigiano Reggiano PDO producers faced problems in keeping product quality standards high. At that time, eleven small dairies from Bibbiano (Reggio Emilia) producing high-quality cheeses based on good pastures for cows and on the skill of local cheesemakers started to show their wares at exhibitions and trade fairs in Italy and abroad [30]. At the end of the 19th century, the Animal and Cheese-making School of Reggio Emilia introduced the innovation of using the whey directly into the production process to cope with cheese quality problems [29]. This innovation is still used nowadays and guarantee to produce cheese without preservatives and the milk pasteurization.

Lines 599 and follow ….this sentence doesn’t read well, it is not clear what this sentence means. The new version is:

The research on the Parmigiano-Reggiano LAFS show acceptable sustainability conditions between 2000 and 2018 and catch the capacity of the system's ability to react to the increasing technological pressure and market competition. the loss of workers along at farm and dairy level is the weakest point of the sustainability of the system that, in long term perspective, can impact also on the image of the product and thus also on the economic sustainability. In a logic of rural development, other jobs opportunity linked to the Parmigiano Reggiano PDO should be created by diversification processes or specific policy strategy. It is difficult to imagine a return to "labour intensity" processing techniques along the value chain, but LAFS, considered as the synergic activity of dairies and territories, might evolves by offering commercial services (such as direct sales at dairy level), or recreational tourism services to their costumers (consumers and tourists), recovering the loss of job position from the Parmigiano Reggiano PDO value chain. This last scenario, although desirable, is not easy to achieve and requires the strategic sharing and collaboration of all the stakeholders that manage the LAFS.
